# Modelling the persistence and control of Rift Valley fever virus in a spatially heterogeneous landscape

Warren S. D. Tennant [1,2✉], Eric Cardinale[3,4], Catherine Côtre-Sossah [3,4], Youssouf Moutroifi[5], Gilles Le Godais[6], Davide Colombi[7], Simon E. F. Spencer [1,8], Mike J. Tildesley [1,2,9], Matt J. Keeling [1,2,9], Onzade Charafouddine[5], Vittoria Colizza [10], W. John Edmunds[11] & Raphaëlle Métras[10,11]

The persistence mechanisms of Rift Valley fever (RVF), a zoonotic arboviral haemorrhagic fever, at both local and broader geographical scales have yet to be fully understood and rigorously quantified. We developed a mathematical metapopulation model describing RVF virus transmission in livestock across the four islands of the Comoros archipelago, accounting for island-specific environments and inter-island animal movements. By fitting our model in a Bayesian framework to 2004–2015 surveillance data, we estimated the importance of environmental drivers and animal movements on disease persistence, and tested the impact of different control scenarios on reducing disease burden throughout the archipelago. Here we report that (i) the archipelago network was able to sustain viral transmission in the absence of explicit disease introduction events after early 2007, (ii) repeated outbreaks during 2004–2020 may have gone under-detected by local surveillance, and (iii) co-ordinated within-island control measures are more effective than between-island animal movement restrictions.

[1] The Zeeman Institute: SBIDER, University of Warwick, Coventry CV4 7AL, UK. [2] Mathematics Institute, University of Warwick, Coventry CV4 7AL, UK. [3] Centre de Coopération Internationale en Recherche Agronomique pour le Développement, UMR Animal, Santé, Territoires, Risques, et Écosystèmes, F-97490Sainte Clotilde, La Réunion, France. [4] Animal, Santé, Territoires, Risques, et Écosystèmes, Université de Montpellier, Centre de Coopération Internationale en Recherche Agronomique pour le Développement, INRAE, Montpellier, France. [5] Vice-Présidence en charge de l'Agriculture, l'Elevage, la Pêche, l'Industrie, l'Energie et l'Artisanat, B.P. 41 Mdé, Moroni, Union of the Comoros. [6] Direction de l'Alimentation, de l'Agriculture et de la Forêt de Mayotte, Service de l'Alimentation, 97600 Mamoudzou, France. [7] Aizoon Technology Consulting, Str. del Lionetto 6, Torino, Italy. [8] Department of Statistics, University of Warwick, Coventry CV4, 7AL, UK. [9] School of Life Sciences, University of Warwick, Coventry CV4 7AL, UK. [10] INSERM, Sorbonne Université, Institut Pierre Louis d'Épidémiologie et de Santé Publique (Unité Mixte de Recherche en Santé 1136), 75012 Paris, France. [11] Centre for the Mathematical Modelling of Infectious Diseases, Department of Infectious Disease Epidemiology, London School of Hygiene and Tropical Medicine, London WC1E 7HT, UK. ✉email: Warren.Tennant@warwick.ac.uk

Rift Valley fever (RVF) is a zoonotic arboviral haemorrhagic fever of increasing global health concern. In most cases, it is asymptomatic in humans, but in some cases, it can cause dengue-like symptoms, or in rare instances, more severe conditions such as meningo-encephalitis, haemorrhagic fever or death. In domestic ruminant livestock (cattle, sheep and goats), RVF virus infections cause waves of abortions and high neonatal deaths[1,2]. RVF was described for the first time in Kenya in 1931[3]. Since then, the disease has been reported throughout Africa, and outside the African continent in Madagascar (1979), in the Arabian Peninsula (2000) and in the Comoros archipelago (2007)[4–6]. Beyond its potential for spread to further geographical areas, a major concern is the likelihood of persistence in previously disease free regions[7–11]. These persistence mechanisms vary between ecosystems depending on local host communities and meteorological factors allowing favourable conditions for mosquito vectors to complete their life cycle and to be capable of virus transmission[12]. Whilst these mechanisms may apply within a geographically limited homogeneous ecosystem, over a larger geographical scale, one needs to account for spatial heterogeneity. This includes considering other factors and mechanisms such as the variability of the environmental conditions impacting vector transmission, or the movements of hosts across space[13–16].

Previous modelling studies for RVF have focused on estimating key transmission mechanisms in single patch systems, e.g. Mayotte[17,18], or on viral spatial spread during or between epidemics, e.g. in South Africa and Uganda[19,20]. However, no study to date has estimated the importance of both environmental variables and animal movements on RVF virus persistence in a spatially heterogeneous system, by fitting a mathematical model to disease data, precluding the formal assessment of disease control measures in a real-world settings. In order to better understand the mechanisms of RVF viral spread and persistence in a spatially heterogeneous system, we developed and fitted a metapopulation model to a series of RVF seroprevalence studies in livestock across the Comoros archipelago—a collection of four islands located in the South-Western Indian Ocean, between Madagascar and Mozambique.

In this paper, we thus sought to (i) estimate the importance of island-specific variables and animal movements across the islands on RVF spread, (ii) assess the likelihood of RVF persistence in the system without re-introduction from mainland Africa or Madagascar, and (iii) assess the impact of livestock movement control measures on disease incidence in the Comoros archipelago. To do this, we developed a mathematical metapopulation model to describe the spread of RVF virus within and between the four islands of the Comoros archipelago (Fig. 1), and fitted this model in a Bayesian framework to livestock seroprevalence data collected from 2004 until 2015. Consequently, we estimated the basic reproduction number of the disease on each island over time, and the mean annual number of livestock which move between the four islands. We then used our model to forecast specific RVF antibody prevalence on all four islands in the absence of another explicit introduction event of the virus from outside the Comoros archipelago. Finally, we assessed the impacts of movement restrictions and reducing within-island transmission on each island upon the total number of new infections in livestock from 2004 to 2015.

## Results

**Livestock seroprevalence data.** We used age-stratified RVF IgG seroprevalence data collected in livestock as part of several sero-surveys conducted amongst the four islands of the archipelago (namely Grande Comore, Mohéli, Anjouan and Mayotte). A total of 8423 samples—2191 in Grande Comore, 475 in Mohéli, 857 in Anjouan and 4900 in Mayotte—were collected over a 12-year period (July 2004–June 2015). Summary statistics for these data are shown in Fig. 2. For details on these data, refer to the Methods section.

**Estimation of island-specific transmission and animal movements.** We modelled RVF viral transmission within age-structured livestock populations within each island as a function of the Normalized Difference Vegetation Index (NDVI), and between islands through the movement of livestock. Five models with different relationships between NDVI and within-island transmission were fitted in a Bayesian framework to the age-stratified livestock seroprevalence data in order to estimate island-specific viral transmission rates. We used Deviance Information Criterion (DIC)[21] to discriminate the relative quality of fitted models (Supplementary Table 1). We present the fit for the tested models in Fig. 2 and Supplementary Figs. 1–4 – showing the comparison between simulated age-stratified IgG seroprevalence in livestock against the available serological data.

The model assuming a similar exponential relationship between NDVI and transmission amongst livestock across islands, with island-specific baseline transmission values (Model 3b) fitted to the empirical data with the greatest accuracy according to DIC (Fig. 2). Predictions of Model 3b included the observed rise in seroprevalence of RVF in Grande Comore and Mohéli and fall in seroprevalence in Anjouan between 2011 and 2014. Furthermore, in the complete absence of age information (Mayotte 2004–2008) the model captured the rise in seroprevalence in 2007–2008. There were only a few serological surveys for which the model was unable to capture. These discrepancies included sero-surveys in young livestock conducted in Grande Comore from April until June 2013.

The full set of parameter estimates for Model 3b can be found in Supplementary Table 2 and Supplementary Fig. 5. These parameter estimates corresponded to a (median) maximum annual seasonal reproduction number, $R_{st}$, for each island. These were 3.99 for Grande Comore (95% credible interval (CrI) = [3.17, 4.72]), 3.40 for Mohéli (95% CrI = [2.33, 5.54]), 2.95 for Anjouan (95% CrI = [2.15, 3.83]) and 2.77 for Mayotte (95% CrI = [2.36, 3.14]). The geometric mean of seasonal reproduction numbers was also greater than one across all four islands (Supplementary Table 3). Consequently, our model inferred multiple outbreaks to have occurred on both Grande Comore and Mohéli: Mohéli in 2011, Grande Comore in 2012 and both Mohéli and Grande Comore in 2014 (Supplementary Fig. 6). In addition, the importation of infectious animals was inferred to begin between December 2006 and April 2007 (with 95% credibility) with 4.15 (95% CrI = [1.33, 7.10]) infectious animals being introduced each week. Livestock were also traded between islands within the Comoros archipelago (Fig. 3). Mayotte was estimated to be the largest importer of animals, with 1875 (95% CrI = [1707, 2058]) importations per annum, all of which came from Anjouan. As a consequence, Anjouan was the largest exporter, exporting 2912 (95% CrI = [2728, 3104]) animals per year, with 629 (95% CrI = [565, 689]) and 408 (95% CrI = [365, 449]) to Grande Comore and Mohéli respectively. Mohéli exported and imported similar number of animals per year: 739 (95% CrI = [685, 795]) imports and 719 (95% CrI = [641, 800]) exports. The majority of Mohéli's exports were trade to Grande Comore, 657 (95% CrI = [588, 726]), with approximately half that imported, 331 (95% CrI = [294, 367]).

**RVF transmission dynamics and persistence in the Comoros archipelago.** Based on 1000 realisations of the best model (Model 3b), we observed a rise in seroprevalence during 2007–2008 on all four islands (Fig. 4). This was due to high seasonal reproduction numbers on each island attributed to high NDVI values

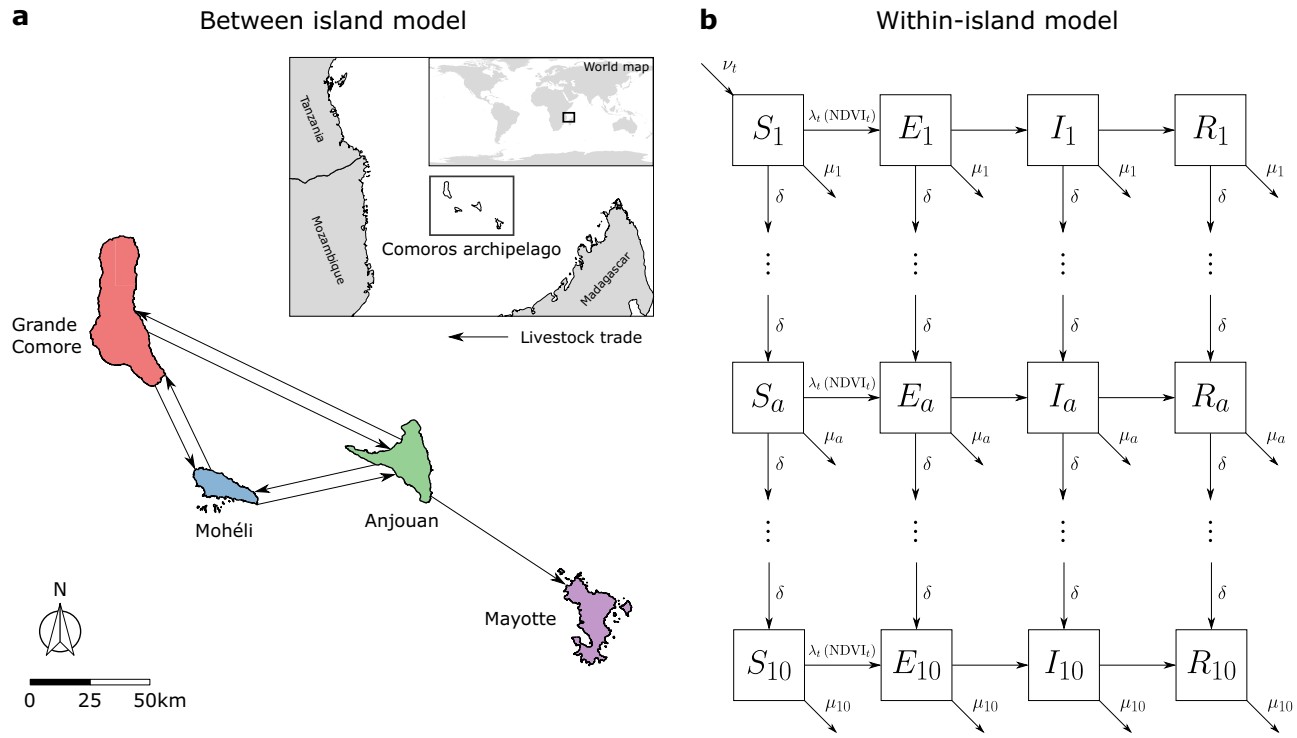

**Fig. 1 Metapopulation model for RVF virus transmission in the Comoros archipelago.** In order to quantify the drivers of Rift Valley fever in the Comoros archipelago, we developed a metapopulation model describing RVF virus infection of livestock (cattle, sheep and goats) in the Comoros archipelago. **a** We modelled the explicit movement of livestock (solid black arrows) between the four islands in the Comoros archipelago: Grande Comore (red), Mohéli (blue), Anjouan (green) and Mayotte (purple). **b** Within-island viral transmission was modelled as an age-stratified Susceptible-Exposed-Infected-Recovered (SEIR) model. The shown schematic illustrates the transfer of animals (arrows) between four infection states–susceptible, $S$, exposed, $E$, infected, $I$, and recovered, $R$–and 10 age groups, $a = 1, ..., 10$, (boxes) on each island. The number of individuals in each compartment was updated in discrete time, $t$, with a single time step equal to one epidemiological week. Animals were born into the youngest age group at a time-varying rate $\nu_t$ and were removed from the system due to death at an age-dependent proportion $\mu_a$. A fixed proportion, $\delta$, of individuals were aged at each time step. Susceptible animals were exposed to the disease at a time-varying proportion, $\lambda_t$, which was dependent on the mean Normalized Difference Vegetation Index (NDVI) across each island and time point. For further details on the metapopulation model, please refer to the Methods section. Data used to produce the maps shown in (**a**) were made available under Attribution 3.0 Unported (CC BY 3.0)[67] and Creative Commons Attribution for Intergovernmental Organisations (CC BY-IGO)[68] licenses. The former (CC BY 3.0) licenced the data for Mayotte[69], and the latter (CC BY-IGO) licenced the data for the Union of Comoros[70], Tanzania[71], Madagascar[72] and Mozambique[73]. All presented data was unaltered.

(Supplementary Fig. 7) occurring alongside a sufficiently high proportion of susceptible animals; or related to the importation of infectious livestock into Grande Comore from the African mainland during late 2006 and early 2007, following the RVF outbreak in East Africa[22].

Forecasting beyond July 2015 until June 2020 showed outbreaks occurring on all four islands again, but their timing and magnitude varied greatly between islands. According to our model predictions, large outbreaks of RVF occurred on Grande Comore and Mohéli in either 2017 and/or 2018. A small outbreak was predicted for Anjouan in 2018, followed by a substantial epidemic in Anjouan and Mayotte during 2019. By the end of June 2020, the model predicted population level seroprevalence to be 28.3% on Grande Comore (95% prediction interval (PI) = [27.0%, 43.1%]), 30.2% on Mohéli (95% PI = [27.7%, 36.6%]), 51.4% on Anjouan (95% PI = [44.0%, 53.9%]) and 20.7% on Mayotte (95% PI = [7.5%, 26.0%]), giving weight to the hypothesis that the Comoros archipelago is able to sustain RVF viral transmission without an explicit introduction of the virus from mainland Africa or Madagascar.

**Impact of livestock movement control measures.** To further investigate the role of animal movements on the epidemiology of

Rift Valley fever in the Comoros archipelago, we compared the total number of livestock infections in the 2004–2015 period under different movement-restriction scenarios (Fig. 5a). Under the full trade network, the estimated number of infections per island were 362,659 (95% CrI = [341,277, 420,691]) in Grande Comore, 57,244 (95% CrI = [50,664, 60,630]) in Mohéli, 98,567 (95% CrI = [91,241, 104,117]) in Anjouan and 7482 (95% CrI = [6,532, 8,527]) in Mayotte.

Movement reductions on Grande Comore generated a median increase of 20,076 in the total number of cases across all four islands compared to the full trade network. Relative to the (median) total number of infections on each island under the full trade network, reducing the number of imports and exports into Grande Comore by 100% increased the median number of infections in Grande Comore itself by 10% (95% CrI = [−4.0%, 18.5%]). The increase in the majority of simulations was due to a delayed 2012–2013 outbreak in Grande Comore, resulting in a small (new) outbreak in 2013–2014 and more severe outbreak in 2014–2015 on the island (Supplementary Fig. 8). Under 100% reduction in Grande Comore's exports, the total number of infections in Mohéli reduced by 12.6% (95% CrI = [10.6%, 18.8%]). Furthermore, the total number of infections in Anjouan and Mayotte decreased by 10% on average under full movement restrictions on Grande Comore.

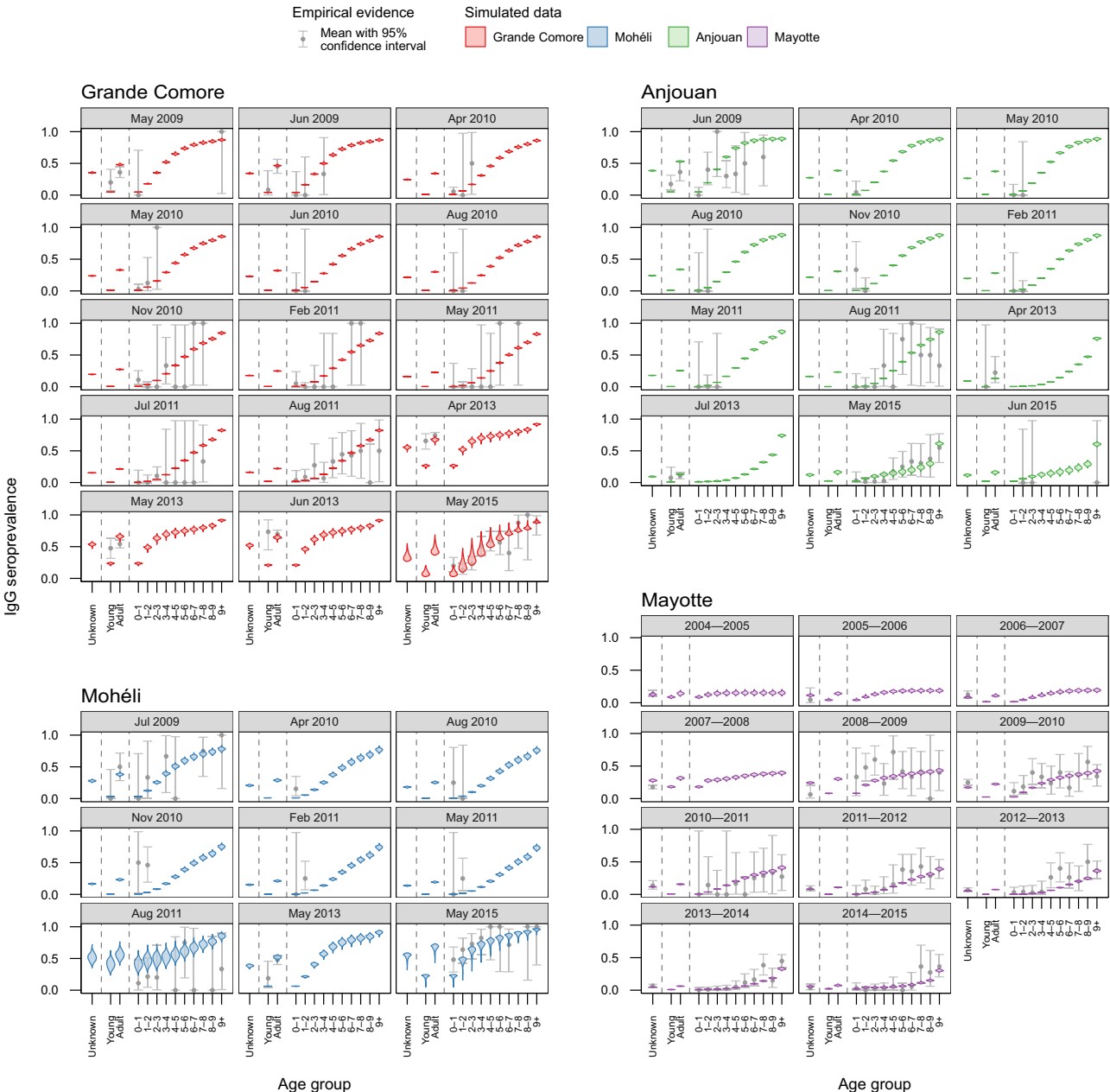

**Fig. 2 Model fit of the best fitted model (Model 3b) to each sero-survey conducted between July 2004 and June 2015.** The exponential transmission model with the same seasonal component $\alpha$ and different baseline transmission $\beta$ for each island fitted to the data best out of all five models tested (DIC = 1189). Shown is the fitted simulated IgG seroprevalence (coloured violins) for each aggregated sero-survey conducted throughout the study period. Seroprevalence of Grande Comore (red), Mohéli (blue) and Anjouan (green) were aggregated by month, and seroprevalence for Mayotte (purple) was aggregated by year. Simulated seroprevalence was generated through 1000 realisations of the metapopulation model. Also shown is the mean observed age-stratified IgG seroprevalence (grey points) with 95% confidence interval (vertical error bars). The presented summary metrics were calculated from the $n = 8423$ biologically independent sera-samples which were used to infer the parameters of the metapopulation model. In particular, there were $n = 2191$ samples for Grande Comore, $n = 475$ samples for Mohéli, $n = 857$ samples for Anjouan and $n = 4900$ samples for Mayotte.

Isolating Mohéli from Grande Comore and Anjouan decreased the (median) total number of infections across all four islands by 15,150. Compared to the full trade network, Mohéli's own total number of infections fell by 14% (95% CrI = [10.0%, 25.8%]). The total number of infections on the other three islands-Grande Comore, Anjouan and Mayotte-were unaffected.

The total number of infections in the Comoros archipelago reduced by 1,998 on average compared to the full trade network under complete movement restrictions to and from Anjouan. Restricting imports and exports of Anjouan by 100%, only reduced its own total number of infections by 14.1% (95%

CrI = [11.6%, 17.3%]), and reduced the total number of infections on Mohéli by ~12.2%.

Restricting movement from Anjouan to Mayotte increased the total number of cases across all four islands on average: a rise of 724 infections over the study period. A complete reduction in imports into Mayotte averted the 2007 Mayotte epidemic, but may have instead caused an outbreak in 2011 owing to sufficient local conditions for transmission (Supplementary Fig. 8). As a result of the movement restrictions to Mayotte, infections on Grande Comore and Mohéli (the first and third most populous islands) increased by 0.5% on average.

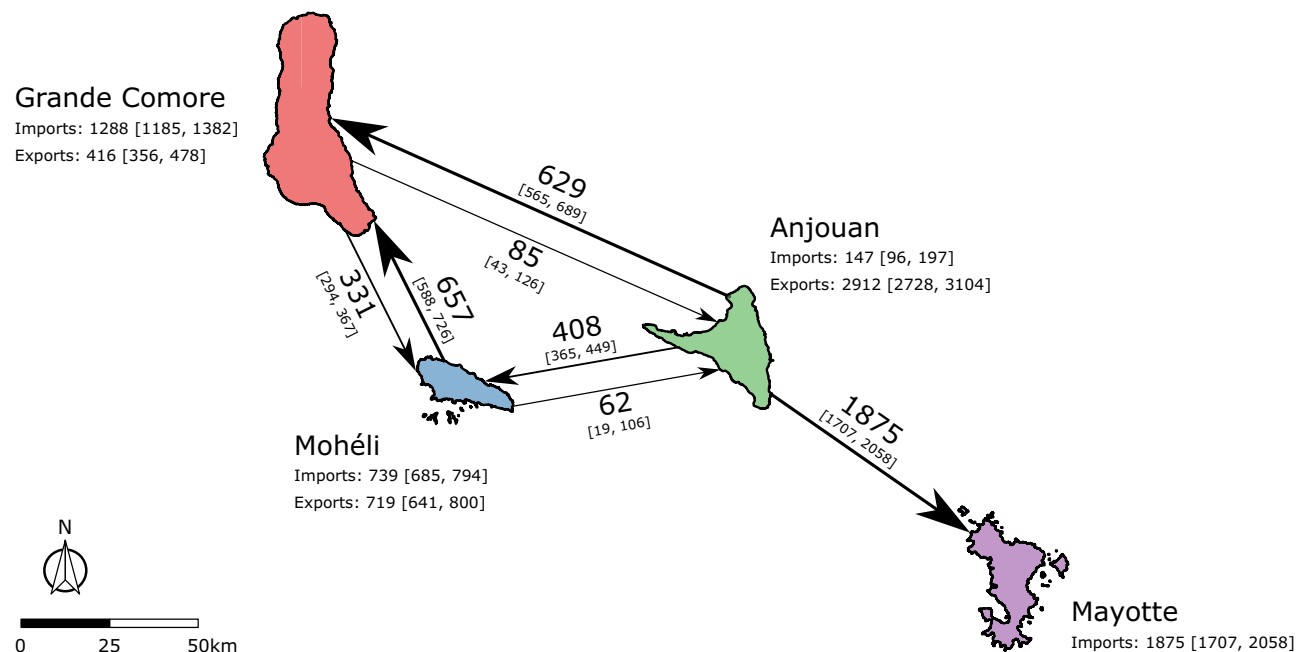

**Fig. 3 Estimated livestock trade network in the Comoros archipelago from best fitted model (Model 3b), presented as the annual number of livestock heads moved between islands.** The imports and exports of each island in the Comoros archipelago—Grande Comore (red), Mohéli (blue) and Anjouan (green) and Mayotte (purple)—were estimated by fitting the metapopulation model to age-stratified sero-surveys conduct from July 2004 until June 2015. The estimated annual trade network of livestock in the Comoros archipelago is shown, with the direction of each arrow indicating the direction of trade between islands. The median and 95% credible interval (CrI) of estimated annual livestock movements are shown on each arrow. Data used to produce the map were made available under Attribution 3.0 Unported (CC BY 3.0)[67] and Creative Commons Attribution for Intergovernmental Organisations (CC BY-IGO)[68] licenses. The former (CC BY 3.0) licenced the data for Mayotte[69], and the latter (CC BY-IGO) licenced the data for the Union of Comoros[70]. All presented data was unaltered.

**Effects of reducing within-island transmission.** To investigate the long-term impacts of island-specific control measures, such as vector control, on the dynamics of RVF throughout the Comoros archipelago, we compared the total number of livestock infections in 2004–2015 period under 10%, 20% and 30% reductions in the transmission rate of each island (Fig. 5b).

Reducing the within-island transmission rate on Grande Comore by 10%, 20% and 30% caused a median decrease in the total number of cases across all four islands by 31,000, 44,882 and 102,693 compared to the full transmission model. These overall decreases in incidence was because of a reduction in the number of cases on Grande Comore itself and Mohéli: decreases of 25% (95% CrI = [15.4, 56.4]) and 12.2% (95% CrI = [10.5%, 20.7%]) on Grande Comore and Mohéli respectively under 30% control. In each scenario, the 2006 and 2012 simulated outbreaks on Grande Comore were not as severe and the 2011 outbreak on Mohéli was delayed until 2013 (Supplementary Fig. 9).

Both a 10% and 30% reduction in the transmission rate on Mohéli reduced the number of cases throughout the Comoros archipelago by 18,626 and 37,632 on average over the study period respectively. However, under a 20% reduction in the transmission rate, the number of cases increased by 54,154 on average. This is because under the 20% scenario, susceptibility is high enough (higher than the full transmission model) for a more severe outbreak to occur on Mohéli in mid-2011 despite the reduction in transmission intensity. This larger outbreak on Mohéli resulted in a larger outbreak in Grande Comore through trade of infected livestock (Supplementary Fig. 10). As a consequence, the total number of cases on Grande Comore increased by 23.5% (95% CrI = [3.5, 28.7]) compared with the full transmission model under 20% control on Mohéli.

The total number of infections in the Comoros archipelago was reduced by 42,280, 69,114 and 82,344 from July 2004 until June

2015 under 10%, 20% and 30% reductions in the transmission rate on Anjouan respectively. Almost all of these reductions occurred on Anjouan: a 98.5% (95% CrI = [97.8, 99.1%]) decrease in the number of cases on Anjouan under 30% control levels. Similar to the movement-restriction scenario on Anjouan, the 2007 Mayotte epidemic did not occur under each control scenario (Supplementary Fig. 11).

Under 10%, 20% and 30% transmission reduction scenarios, the number of cases on Mayotte decreased by 51.1% (95% CrI = [44.3%, 57.3%]), 75.0% (95% CrI = [71.5%, 78.3%]) and 86.5% (95% CrI = [84.7%, 88.3%]) respectively. The temporal epidemiological dynamics of the other three islands were unaffected under control on Mayotte (Supplementary Fig. 12).

## Discussion

Understanding the transmission dynamics of Rift Valley fever (RVF) within animal populations is essential towards estimating human spillover risk and assessing the impact of control measures[23,24]. Characterising persistence mechanisms for RVF are useful to assist long-term surveillance programmes, anticipate re-emergence and assess the impact of control measures[25]. In spatially heterogeneous systems, these persistence mechanisms not only include factors at a local scale, but also those over a larger geographical scale, such as pathogen re-introduction from neighbouring regions[26]. However, the effects of hosts, vectors and their environment on the persistence of RVF are not yet well understood or quantified. In order to improve understanding of how environmental factors and animal trade influence the transmission dynamics of Rift Valley fever, we developed a metapopulation model for RVF infection in livestock and fitted it to data in a multi-insular ecosystem – the Comoros archipelago – from serological data during 2004–2015.

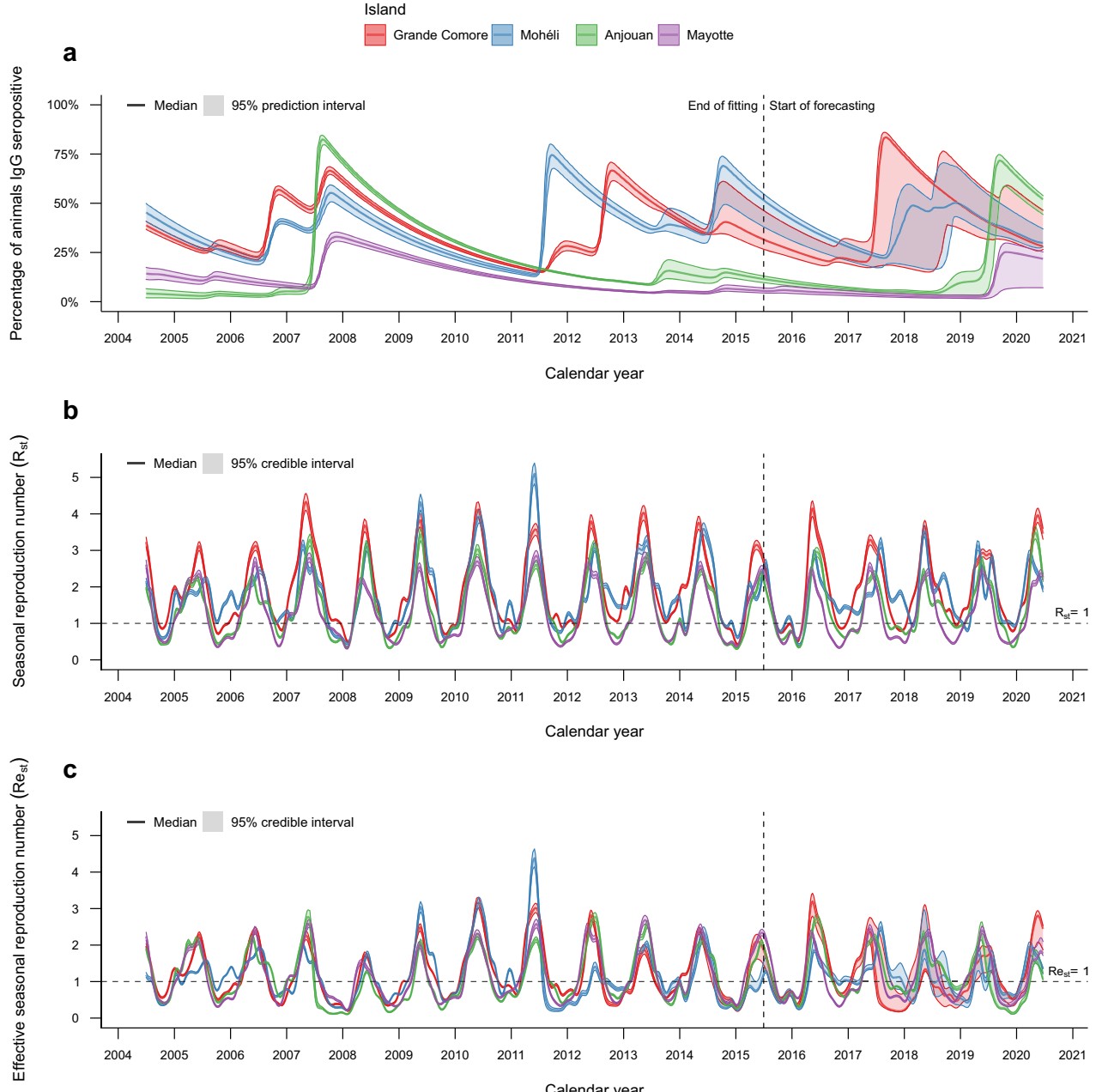

**Fig. 4 Simulated RVF IgG seroprevalence on each island in the Comoros archipelago, 2004–2020, for the best fitted model (Model 3b).** The metapopulation model was fitted to age-stratified sero-surveys from July 2004 until June 2015 for Grande Comore (red), Mohéli (blue), Anjouan (green) and Mayotte (purple). Fitted models were then simulated until July 2020. **a** Median (solid line) and 95% prediction interval (coloured bands) of IgG seroprevalence. **b** Median (solid line) and 95% credible interval (coloured bands) of the seasonal reproduction number ($R_{st}$). **c** Median (solid line) and 95% credible interval (coloured bands) of the effective seasonal reproduction number ($Re_{st}$). Distributions shown were estimated from 1000 realisations of the metapopulation model.

We showed that (i) the within-island RVF virus transmission was similarly driven by NDVI across these islands, (ii) the virus was able to persist across the network over 12-year period, (iii) within-island controls were more effective than livestock movement restrictions between islands. Finally, we provided evidence that some recent outbreaks may have gone undetected.

In total, five different metapopulation models were fitted: each with their own model describing the rate of transmission between livestock. The transmission mechanisms mapping our seasonal driver, NDVI, to transmission rate in an exponential manner fitted to the data best according to Deviance Information Criterion (DIC). This result not only agrees with an NDVI-driven

transmission model for RVF applied to Mayotte only[17], but suggests that the effects of our choice of covariate (NDVI), alongside island-specific baseline transmission rates, were sufficient to explain the observed serology on each island. This demonstrates that NDVI can be successfully used as a proxy for the effects of seasonal mosquito population dynamics on the transmission, and thus may be applied to other arboviral disease systems where entomological data is scarce. However, it is still important to evaluate how other factors might influence baseline transmission rates. In our study, the estimated baseline transmission rates on each island from highest to lowest were Grande Comore, Mohéli, Anjouan and Mayotte. This variation may be

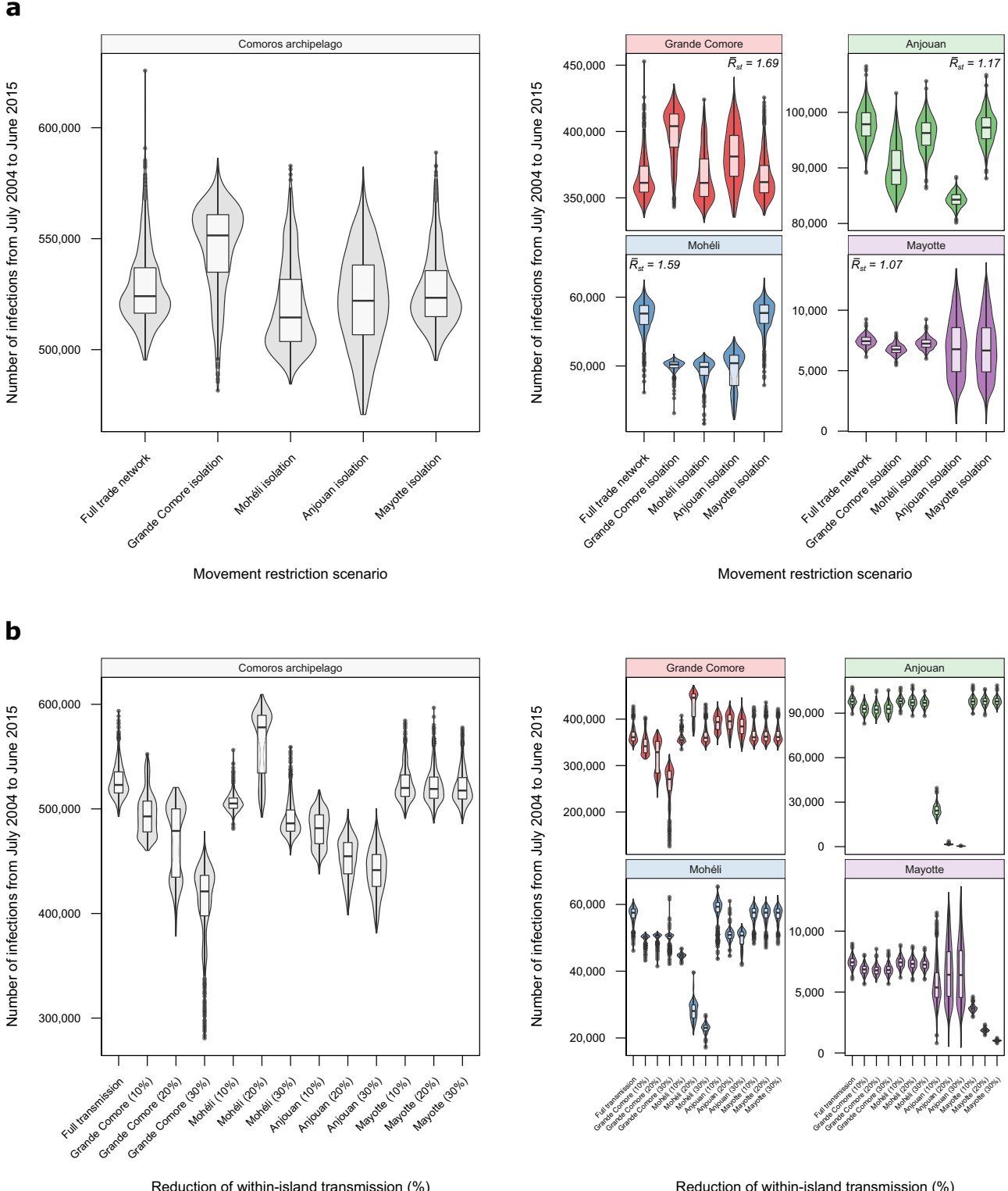

**Fig. 5 Effect of control measures on the number of infections in the Comoros archipelago from 2004 to 2015.** The total number of infections from July 2004 until June 2015 is shown under each control measure scenario on each island in the Comoros archipelago (grey): Grande Comore (red), Mohéli (blue), Anjouan (green) and Mayotte (purple). **a** Full (100%) restrictions on imports and exports were placed on each island. For each island, the median estimate of the geometric mean seasonal reproduction number, $\bar{R}_{st}$, independent of movement restrictions is shown. **b** Within-island transmission rate was reduced by 10%, 20% and 30% on each island independently. The shown violins and boxplots were calculated from $n = 1000$ independent realisations of each control scenario. The central line of the boxplots defines the median with the lower and upper bounds of the box corresponding to the first and third quartiles. The upper whisker extends from the upper bound of the box to the largest value no further than 1.5 × IQR from the upper bound (where IQR is the inter-quartile range). Similarly, the lower whisker extends from the lower bound of the box to the smallest value at most 1.5 × IQR of the lower bound. Data beyond the end of the whiskers are outlying points that are plotted individually.

attributed to the different agricultural ecosystems on each island, such as livestock production system or within-island trade network of varying intensity[27–29]. These findings were reflected in the (time-varying) reproduction number: a metric to quantify disease severity and threshold criterion to determine endemicity[30].

Across all four islands, the geometric mean seasonal reproduction number was greater than one, indicating persistence of RVF virus in the Comoros archipelago. The maximum estimated reproduction number for each island was between 2.5 and 4, which is in line with previous reproduction number estimates for RVF[20,31,32]. The seasonal reproduction number was greater than one for over three quarters of the year on Grande Comore and Mohéli, and only half the year on Anjouan and Mayotte (Supplementary Table 3), reflecting island-specific conditions. Lower reproduction number estimates on Anjouan and Mayotte might explain why outbreaks were less likely to occur on these islands, as time periods with conditions suitable for RVF transmission (high NDVI) were offset by time periods with poor transmission suitability (low NDVI). This explanation at least agrees with existing evidence that an explicit introduction event from outside the Comoros archipelago may have been essential for an outbreak to occur on Mayotte in 2006/2007[5,6,17].

Importation of RVF infected animals from outside the Comoros archipelago may have also been an essential factor in causing the 2018 epidemic in Mayotte[18,33]. In the absence of an explicit re-introduction event of the RVF virus in 2017/18, our model predicted that an outbreak was ongoing in Mayotte 1 year after serological data suggests. Indeed, sequence analysis of the strains confirmed that the strain circulating in Mayotte from 2018 onwards was that responsible for the RVF epidemics in Uganda in 2017[34]. Yet even without an explicit re-introduction of the virus into Grande Comore within our framework, our model predictions suggest that the environmental conditions of Grande Comore and Mohéli were sufficient to cause substantial RVF outbreaks in 2017 and 2018. There is little empirical evidence to support these predictions as it stands, owing to the lack of recent active surveillance on these islands, suggesting these outbreaks may have been missed. As a consequence, any vaccination campaign that would be reactive to the detection of outbreaks would currently be ineffective unless surveillance was first improved.

Our study showed that restricting imports into Mayotte prevented an epidemic in 2007 (Supplementary Fig. 8). This suggests that at least for Mayotte, the importation of livestock from Anjouan to Mayotte is paramount for inducing large outbreaks on the island. Indeed, viral re-emergence on Mayotte resulted from viral re-introductions (likely from infected animals imported from neighbouring islands), coupled with an important proportion of the local livestock being susceptible to infection[17,18,35]. However, in contrast to previous data that showed Mayotte by itself in a closed ecosystem (without animal imports) could not sustain viral transmission between its two epidemics[8,17], our results indicated that an outbreak may have occurred during 2011 on Mayotte instead due to sufficient environmental conditions for transmission.

We demonstrated that island-specific control strategies, such as movement restrictions or vector control, may result in more poor epidemiological outcomes over the long term. For example, restricting livestock movements to and from Grande Comore only served to delay an outbreak to a season which was more suitable for transmission in the majority of simulations, resulting in a more severe outbreak. However, <5% of movement-restriction control simulations caused a decrease to the total number of infections Grande Comore, suggesting that there may be scenarios in which movement-restriction control measures might be effective. As Grande Comore was also the island least

affected by the control measures of the other three islands, our results may suggest that only within-island control (perhaps with some combination of movement restrictions) ought to be considered to reduce disease burden on Grande Comore. However, Mohéli's temporal dynamics were found to be very sensitive to controls implemented on Grande Comore or Anjouan. Island-specific controls on Mayotte affected the other three islands the least, and thus only Mayotte is appropriate for focused island-specific control measures. A combination of the controls may be more appropriate to implement on Grande Comore, Mohéli and Anjouan simultaneously, but the potential impacts on the complete system (including Mayotte) should be thoroughly assessed first.

There are some limitations to using a mechanistic approach to elucidate the factors that drive spread and persistence of RVF. Although we showed that the disease continued to propagate without an explicit introduction event from continental Africa after 2007, this may be attributed to our choice of modelling framework: a deterministic framework in which disease can technically never be eradicated. That is, our choice of framework does not capture additional uncertainties associated with random infection processes, which may result in stochastic extinction of the disease. However, rather than discussing persistence as an absence of stochastic extinction, we have presented our claims of persistence in terms of the reproduction number. Our results showed that, on average, RVF has the ability to remain endemic on the Comoros archipelago provided that the climate conditions remain favourable to support transmission. Within our framework, implicit persistence mechanisms can be ascribed to alternate wildlife hosts[36], irregular introductions of infected hosts through trade[15], or maintenance of the virus in local mosquito populations via transovarial transmission[37,38]. Serological surveys in potential wildlife hosts, imported animals and mosquito vectors would enable the routes for RVF persistence to be more precisely assessed. Within our metapopulation model we assumed that movements of animals between islands were constant over time due to lack of data. These movement patterns had only been previously described qualitatively and suggest that they depend on several factors including climate, economic reasons, religious festivities as well as weddings and other family gatherings[6,27]. These seasonal oscillations in host movement may have important implications on the spread of the RVF across the Comoros archipelago[39,40]. With quantitative, time-dependent information on livestock movement, our framework could be adapted to include these seasonal movement patterns. We also did not explicitly model mosquitoes within our framework as there were insufficient data on all competent vectors of RVF in the Comoros archipelago, or elsewhere, to parametrise such an approach[27,41,42]. It is clear however that understanding mosquito population dynamics and their vectorial capacity for RVF virus will be paramount to fully disentangle the seasonal drivers of RVF from one another. Our model could then be extended to include the population dynamics of RVF vectors and wildlife hosts in areas where such data are available.

Despite these limitations, our modelling framework and parameter estimates might be adapted to settings within and outside of RVF and/or the Comoros archipelago. Our transmission parameter estimates could serve as initial inputs to calibrate modelling approaches to other affected regions, such as Kenya[43], Tanzania[44] and South Africa[45,46]. The presented modelling framework and/or livestock movement estimates may also be adapted to other ruminant diseases that have been shown to circulate in the Comoros archipelago, such as Q-fever[47] and ovine rinderpest (also known as *peste des petits ruminants*)[48]. With increasingly fine-scale serological data, livestock populations may further be spatially refined to include multiple communities per

island, with transmission between communities described using a coupled-population approach[49]. Transmission parameters of each community could then be estimated within a hierarchical structure, whereby transmission drivers between neighbouring communities may be similar. These serological data may further be used to infer these host movement dynamics in scenarios where prior estimates are uninformative (Supplementary Fig. 13). This approach could thus be used to quantify movements of birds or bats, for example, between different communities in multi-insular systems[50]. These proposed model extensions and applications however hinge on the appropriate demographic and epidemiological data to be readily available and carefully maintained alongside local expert knowledge of potential disease transmission pathways.

In summary, we have presented a metapopulation model for RVF fitted to empirical data. We have shown that the virus is able to persist in the Comoros archipelago without the need for the explicit large-scale introduction of the virus from eastern Africa or Madagascar. Moreover, we have identified that the ecological features of Grande Comore and Mohéli were more suitable to maintain the virus, whereas livestock trade and high susceptibility were essential for RVF epidemics to occur on Anjouan and Mayotte. Our results suggest that several outbreaks occurred in Grande Comore and Mohéli that were missed owing to insufficient surveillance, emphasising the importance of sustaining long-term, co-ordinated surveillance programmes in order to elicit an early enough control response to avert epidemics in livestock and resultant spillover into human populations. The presented metapopulation model and parameter estimates for disease transmission and hosts movements, may also be extended to other settings within and outside of Rift Valley fever.

## Methods

**Study area**. The Comoros archipelago is a group of four islands—Grande Comore (1146 km$^2$), Mohéli (290 km$^2$), Anjouan (424 km$^2$) and Mayotte (374 km$^2$)—located in the northern part of the Mozambique Channel, between Mozambique and Madagascar, populated with ~1 million inhabitants[29,51]. The climate of the islands is marine tropical, and the Comoros archipelago are old volcanic islands, with varying ecosystems[29,50,52]. The livestock (cattle, sheep and goat) production system is extensive, and the total animal population is estimated to be over 350,000[53]. Animals are raised for local consumption. Grande Comore, Mohéli and Anjouan are part of the Union of the Comoros, and may exchange animals regularly. Mayotte has been a French department since 2011 and a EU outermost territory since 2014, and no official import of animals are reported from the neighbouring island, Anjouan, whilst some unreported imports occur on a regular basis[12,27,50,52].

**Livestock seroprevalence data**. We analysed cross-sectional seroprevalence data conducted in the Union of Comoros as part of surveillance programmes in Grande Comore, Mohéli and Anjouan as presented by Roger et al.[54] and Roger et al.[27]. Surveys were conducted in 2009, 2012, 2013 and 2015. The livestock prevalence data from Mayotte covered the period 2004 to 2016 and resulted from the SESAM (Système d'épidémio-surveillance animale à Mayotte) surveillance system as presented in Métras et al.[12] and Métras et al.[17].

**Metapopulation model**. To decipher the relative roles of meteorological factors and livestock movements on RVF persistence across the four islands of the Comoros archipelago, we developed a metapopulation model, using an island as a patch. Within each of the four patches, we modelled RVF transmission dynamics and islands were connected by animal movements. A schematic summarising the main components of the model is shown in Fig. 1.

The model was deterministic and discrete-time, where each time step was 1 week. The livestock populations (cattle, sheep and goats) were modelled for each island in the Comoros archipelago, with each split into 10 age groups $a$: 0–1 ($a = 1$), 1–2 ($a = 2$), ..., and 9+ ($a = 10$) years old. At time point $t$, each island $i$ contained $N_{t,i,a}$ animals of each age group $a$, and an age-dependent proportion of animals, $\mu_a$, died at each time step. Animals of each age group $a$ moved from island $i$ to island $j$ at a weekly rate $m_{t,ij,a}$, where $m_{t,ii,a}$ denoted the number of animals of age group $a$ which remained on island $i$ at time $t$. Animals were born into the youngest age group at rate $\nu_{t,i}$ on each island $i$ and a proportion of animals, $\delta$, were aged at each time step. For simplicity, we assumed that the total livestock population of each island was constant over time.

Animals of each age group $a$ and island $i$ were classified as either susceptible ($S_{i,a}$), exposed ($E_{i,a}$), infectious ($I_{i,a}$) or recovered ($R_{i,a}$) to the virus. We assumed that animals were exposed to the virus for one week, and remained infectious for 1 week. Once recovered, animals were immune to infection for the duration of their life. We assumed no maternal protection to the virus, and thus new animals were assumed to be fully susceptible to the disease. At each time $t$, an island-dependent proportion of susceptible animals became infected, denoted $\lambda_{t,i}$. Infectious animals of age $a$ were introduced into each island $i$ from sources outside the Comoros archipelago at a time-dependent rate denoted by $I_{t,i,a}^{(ext)}$.

Given livestock population sizes at time $t$, the total number of susceptible, exposed, infectious and recovered livestock for each age group $a$ and island was calculated at subsequent time $t + 1$ as follows:

For animals less than one year old (the first age group, $a = 1$):

$$S_{t+1,i,1} = \nu_{t,i} + \sum_{j=1}^{4} \frac{m_{ji,1}}{N_{t,j,1}} \left[ (1 - \delta)(1 - \mu_1)\left(1 - \lambda_{t,j}\right)S_{t,j,1} \right], \quad (1)$$

$$E_{t+1,i,1} = \sum_{j=1}^{4} \frac{m_{ji,1}}{N_{t,j,1}} \left[ (1 - \delta)(1 - \mu_1)\lambda_{t,j}S_{t,j,1} \right], \quad (2)$$

$$I_{t+1,i,1} = \sum_{j=1}^{4} \frac{m_{ji,1}}{N_{t,j,1}} \left[ (1 - \delta)(1 - \mu_1)E_{t,j,1} \right] + I_{t,i,1}^{(ext)}, \quad (3)$$

$$R_{t+1,i,1} = \sum_{j=1}^{4} \frac{m_{ji,1}}{N_{t,j,1}} \left[ (1 - \delta)(1 - \mu_1)\left(I_{t,j,1} + R_{t,j,1}\right) \right]. \quad (4)$$

For animals between 1 and 9 years old (age groups $a = 2, ..., 9$):

$$S_{t+1,i,a} = \sum_{j=1}^{4} \frac{m_{ji,a}}{N_{t,j,a}} \left[ (1 - \delta)(1 - \mu_a)\left(1 - \lambda_{t,j}\right)S_{t,j,a} \right] + \sum_{j=1}^{4} \frac{m_{ji,a-1}}{N_{t,j,a-1}} \left[ \delta(1 - \mu_{a-1})\left(1 - \lambda_{t,j}\right)S_{t,j,a-1} \right], \quad (5)$$

$$E_{t+1,i,a} = \sum_{j=1}^{4} \left\{ \frac{m_{ji,a}}{N_{t,j,a}} \left[ (1 - \delta)(1 - \mu_a)\lambda_{t,j}S_{t,j,a} \right] + \frac{m_{ji,a-1}}{N_{t,j,a-1}} \left[ \delta(1 - \mu_{a-1})\lambda_{t,j}S_{t,j,a-1} \right] \right\}, \quad (6)$$

$$I_{t+1,i,a} = \sum_{j=1}^{4} \left\{ \frac{m_{ji,a}}{N_{t,j,a}} \left[ (1 - \delta)(1 - \mu_a)E_{t,j,a} \right] + \frac{m_{ji,a-1}}{N_{t,j,a-1}} \left[ \delta(1 - \mu_{a-1})E_{t,j,a-1} \right] \right\} + I_{t,i,a}^{(ext)}, \quad (7)$$

$$R_{t+1,i,a} = \sum_{j=1}^{4} \frac{m_{ji,a}}{N_{t,j,a}} \left[ (1 - \delta)(1 - \mu_a)\left(I_{t,j,a} + R_{t,j,a}\right) \right] + \sum_{j=1}^{4} \frac{m_{ji,a-1}}{N_{t,j,a-1}} \left[ \delta(1 - \mu_{a-1})\left(I_{t,j,a-1} + R_{t,j,a-1}\right) \right]. \quad (8)$$

For animals <9 years old (the final age group, $a = 10$):

$$S_{t+1,i,10} = \sum_{j=1}^{4} \left\{ \frac{m_{ji,10}}{N_{t,j,10}} \left[ (1 - \mu_{10})\left(1 - \lambda_{t,j}\right)S_{t,j,10} \right] + \frac{m_{ji,9}}{N_{t,j,9}} \left[ \delta(1 - \mu_9)\left(1 - \lambda_{t,j}\right)S_{t,j,9} \right] \right\}, \quad (9)$$

$$E_{t+1,i,10} = \sum_{j=1}^{4} \left\{ \frac{m_{ji,10}}{N_{t,j,10}} \left[ (1 - \mu_{10})\lambda_{t,j}S_{t,j,10} \right] + \frac{m_{ji,9}}{N_{t,j,9}} \left[ \delta(1 - \mu_9)\lambda_{t,j}S_{t,j,9} \right] \right\}, \quad (10)$$

$$I_{t+1,i,10} = \sum_{j=1}^{4} \left\{ \frac{m_{ji,10}}{N_{t,j,10}} \left[ (1 - \mu_{10})E_{t,j,10} \right] + \frac{m_{ji,9}}{N_{t,j,9}} \left[ \delta(1 - \mu_9)E_{t,j,9} \right] \right\} + I_{t,i,10}^{(ext)}, \quad (11)$$

$$R_{t+1,i,10} = \sum_{j=1}^{4} \left\{ \frac{m_{ji,10}}{N_{t,j,10}} \left[ (1 - \mu_{10})\left(I_{t,j,10} + R_{t,j,10}\right) \right] + \frac{m_{ji,9}}{N_{t,j,9}} \left[ \delta(1 - \mu_9)\left(I_{t,j,9} + R_{t,j,9}\right) \right] \right\}. \quad (12)$$

At each time $t$, the birth rate, $\nu_{t,i}$, and force of infection, $\lambda_{t,i}$, were defined as:

$$\nu_{t,i} = \sum_{a=1}^{10} \sum_{j=1}^{4} \frac{m_{ji,a}}{N_{t,j,a}} \mu_a N_{t,i,a} - \sum_{a=1}^{10} I_{t,i,a}^{(ext)}, \quad (13)$$

$$\lambda_{t,i} = 1 - \exp\left(-b_{t,i}\frac{I_{t,i,a}}{N_i}\right), \quad (14)$$

for any island $i$ and time-dependent transmission rate $b_{t,i}$, where $N_i = \sum_{a=1}^{10} N_{t,i,a}$.

At the first time point ($t = 0$), a small number of exposed and infectious animals were initialised on each island (Table 1). A proportion of the total population on each island, $\epsilon_i$, was also assumed to be immune at the start of the simulation. It was also assumed that the age of immune animals was proportional to the age of the population. Therefore, for each island $i$,

$$S_{0,i,a} = \left(N_i - E_{0,i} - I_{0,i} - \epsilon_i N_i\right)p_a, \quad (15)$$

**Table 1 Fixed model parameters.**

| Parameter | Description | Value |
|---|---|---|
| $\delta$ | Proportion of animals ageing each week | 1/48 |
| $\mu_a$ | Proportion of animals in age group $a \in [1, 9]$ dying each week | $8.8 \times 10^{-3}$ |
| $\mu_{10}$ | Proportion of animals in age group 10 dying each week | $6.2 \times 10^{-3}$ |
| $E_{0,i}$ | Initial number of exposed livestock on island $i$ | 5 |
| $I_{0,i}$ | Initial number of infectious livestock on island $i$ | 5 |
| $N_1$ | Total population on Grande Comore | 224,353 |
| $N_2$ | Total population on Mohéli | 31,872 |
| $N_3$ | Total population on Anjouan | 93,616 |
| $N_4$ | Total population on Mayotte | 20,052 |

The parameters in the metapopulation model were chosen based on the current understanding of the epidemiology of Rift Valley fever and the demography of the livestock population in the Comoros archipelago. The remaining parameters were estimated by fitting the model to the serological data collected between 2004 and 2015.

$$E_{0,i,a} = E_{0,i}p_a, \tag{16}$$

$$I_{0,i,a} = I_{0,i}p_a, \tag{17}$$

$$R_{0,i,a} = \epsilon_i N_i p_a, \tag{18}$$

where $p_a$ denotes the proportion of the livestock population in age-group $a$.

*Movement between islands.* We assumed that the total number of movements between islands was constant over time. Owing to the substantial distances between each island, we assumed that animals could move along the network motivated by Roger et al.[27]. The movement network is shown in Fig. 1. That is, only the following movements were possible: Grande Comore to Mohéli, Grande Comore to Anjouan, Mohéli to Grande Comore, Mohéli to Anjouan, Anjouan to Grande Comore, Anjouan to Mohéli, and Anjouan to Mayotte. All animals that did not move on this network were assumed to remain on their respective islands.

As the size of adult livestock relative to the size of boats used to travel between islands was large, only the two youngest age groups were moved between islands. We also assumed that the number of movements per age group $a$ was proportional to the total number of animals in each age group at time $t$ and the total number of weekly movements between island $i$ and island $j$.

$$m_{t,ij,a} = \begin{cases} m_{ij}\dfrac{N_{t,i,a}}{N_{t,i,1}+N_{t,i,2}}, & \text{for } a \in \{1,2\}, \\ 0, & \text{otherwise,} \end{cases} \tag{19}$$

where $m_{ij}$ denoted the total number of weekly livestock movements between island $i$ and island $j$.

*External importation of infection.* On account of the RVF outbreak in eastern Africa between 2006 and 2007, we assumed that infectious animals were imported into Grande Comore for a limited time. As with movements between islands in the Comoros archipelago, we assumed that only the two youngest age groups were imported. We also assumed that the age of imported animals was proportional to the age of the Comoros archipelago livestock population. Given the weekly number of infectious animals imported, $I_{t,i}^{(\text{ext})}$, the number of infectious imports per island $i$ and age group $a$ was defined as follows:

$$I_{t,i,a}^{(\text{ext})} = \begin{cases} I_{t,i}^{(\text{ext})}\dfrac{p_a}{p_1+p_2} & \text{for } t_{\text{start}}^{(\text{ext})} \leq t \leq t_{\text{start}}^{(\text{ext})} + t_{\text{duration}}^{(\text{ext})}, a \in \{1,2\} \text{ and } i = 1, \\ 0 & \text{otherwise,} \end{cases} \tag{20}$$

where $t_{\text{start}}^{(\text{ext})}$ and $t_{\text{end}}^{(\text{ext})}$ denote the start and end of the importation window respectively.

*Seasonally driven transmission functions.* We used NDVI as a proxy for the effects of seasonal mosquito population dynamics on RVF transmission rates. The estimated NDVI at each time point and island was obtained by aggregating NDVI estimates of 250m by 250m grid squares for each island[55]. Weekly estimates were obtained by smoothing the aggregated estimates with a Gaussian kernel of 21 days[56]. As the relationship between NDVI and transmission rate is unknown, we tested three underlying models for transmission: linear, exponential and constant as presented below.
Linear: the transmission rate scaled linearly with NDVI:

$$b_{t,i} = \alpha_i\Big(\text{NDVI}_{t,i} - \min_\tau \text{NDVI}_{\tau,i}\Big) + \beta_i, \tag{21}$$

for some positive parameter $\alpha_i$ and $\beta_i$ representing the baseline level of transmission on each island.

Exponential: the transmission rate scaled exponentially with NDVI:

$$b_{t,i} = \exp\Big[\alpha_i\Big(\text{NDVI}_{t,i} - \min_\tau \text{NDVI}_{\tau,i}\Big) + \beta_i\Big], \tag{22}$$

for some positive parameter $\alpha_i$ and $\exp(\beta_i)$ denoting the baseline level of transmission on each island.
Constant: as a baseline comparison, we also modelled a time-independent transmission rate, thus not depending on NDVI:

$$b_{t,i} = \beta_i, \tag{23}$$

where $\beta_i$ denotes the island-dependent transmission rate.
As the exposure and infectious periods were fixed at one week, the seasonal reproduction number and effective seasonal reproduction numbers for each island were given by $R_{st,i} = b_{t,i}$ and $Re_{st,i} = \frac{R_{t,i}}{N_i}b_{t,i}$ respectively. An island $i$ was deemed to allow the virus to persist if the geometric mean of $R_{s_t,i}$ over the time period was greater than 1.

**Parameters.** The parameters of the model were selected based on the current knowledge on RVF epidemiology and demography of the livestock population in the Comoros archipelago. The parameters that were fixed in the model are shown in Table 1. To allow for monthly aggregates of RVF IgG seroprevalence to be calculated (used in model fitting), the time steps of the model represent 1.08 calendar weeks. Therefore, the weekly ageing rate $\delta$ was chosen such that animals took 48 time steps (1 year) to age. The mortality rates of age groups 1–9 and 10 were set to $8.8 \times 10^{-3}$ and $6.2 \times 10^{-3}$ respectively[57,58]. We assumed that mortality rates of livestock were independent of the island on which they were kept. The population size of each island was calculated by aggregating estimates of sheep, goat and cattle population estimates for each island using the Gridded Livestock Map of the World[53]. All other parameters (Supplementary Table 4) were estimated by fitting the model to serological surveys carried out on each island from July 2004 until June 2015.

**Model fitting and parameter estimation.** To estimate the remaining parameters $\theta$ of the model (Supplementary Table 4), we fitted the metapopulation model to the serological data in a Bayesian framework. Below, we define the observation model (likelihood) and priors of each parameter that were used to fit our model to the livestock seroprevalence data in this framework.

*Observation model.* As farms and animals were randomly sampled for serological testing on each island, we assumed that the number of RVF specific IgG antibody positive animals was binomially distributed given the total number of animals tested and the proportion of the livestock population that were immune at the time of testing. For Grande Comore, Mohéli and Anjouan, surveys were aggregated by the month in which they were conducted. As surveys in Mayotte were conducted throughout the year, we aggregated surveys by epidemiological year, which began in July. Simulating the metapopulation model forward in time to estimate the proportion of the population that were immune at the age of testing,

$$y_{T,i,A}^{(\text{pos})} \sim \text{Binom}\left(n_{T,i,A}^{(\text{tested})}, \frac{\sum_{t\in T}\sum_{a\in A}R_{t,i,a}}{\sum_{t\in T}\sum_{a\in A}N_{t,i,a}}\right), \tag{24}$$

where $y_{T,i,A}^{(\text{pos})}$ and $n_{T,i,A}^{(\text{tested})}$ denote the number of animals that were bled and tested RVF antibody positive in age group(s) $A$ on island $i$ over the aggregated time window $T$, respectively. For surveys that were not age-specific, animals were classed as either adults (age groups 2–10) and infants (age group 1). In surveys where the age of animals was unknown, all age groups in the model were used to calculate the proportion immune at the time of testing.

As each survey was conducted independently from one another, the likelihood of jointly observing all serological surveys was given by the product of observing each survey individually.

*Prior distribution.* The priors for each parameter estimated are shown in Supplementary Table 4. Informative normal priors were chosen for the movement parameters, with prior parameters selected based on consultation with the Comorian veterinary services and previously reported inter-island trade estimates[17,27]. The priors for the proportion of the livestock population immune on each island was set to be a beta distribution with mean seroprevalence of 10% on Anjouan and Mayotte, and 40% on Grande Comore and Mohéli[54]. The variance of each beta distribution was determined after consultation with local experts. Weak normal priors were placed on transmission parameters as these are yet to be robustly quantified. Informative normal priors were used for the start and duration of the import into Grande Comore, with prior parameters selected to correspond to reports of the RVF outbreak in Kenya during 2006/7[59,60]. An weakly informative normal prior was used for the size of the import as there is scarce information on infectivity of imported animals during 2006/2007.

*Posterior distribution.* With the definition of the likelihood and priors for parameters, we sampled from the posterior distribution of the parameters $\theta$, using an adaptive Markov Chain Monte Carlo Metropolis-Hastings random walk algorithm[61,62].

This algorithm constructed a Markov chain which converged to the desired posterior distribution. It did this by, at step $i$ in the chain, first proposing a set of candidate parameters $\theta'$ from a proposal distribution $q$. These candidate parameters were then used to simulate the metapopulation model forward in time. Following this, model output was used to calculate the likelihood of observing the serological data given the observation model described above. These candidate parameters were accepted ($\theta_i := \theta'$) or rejected ($\theta_i := \theta_{i-1}$) with acceptance ratio:

$$\frac{\pi(y|\theta')\pi(\theta')q(\theta'|\theta_{i-1})}{\pi(y|\theta_{i-1})\pi(\theta_{i-1})q(\theta_{i-1}|\theta')}, \tag{25}$$

where $\pi(y|\theta)$ denotes the likelihood of observing all data $y$ with model parameters $\theta$, and $\pi(\theta)$ is the prior of $\theta$.

The chain was initialised with parameters randomly sampled from their respective prior distribution. Candidate parameters were proposed jointly at each step $i$ using a multivariate normal distribution with mean $\theta_{i-1}$ and covariance matrix $(2.38)^2\Sigma_{i-1}/d$, where $\Sigma_{i-1}$ was the empirical covariance matrix of the parameter chains up to iteration $i-1$, and $d$ was the number of proposed parameters[61]. Markov chains of 1,000,000 values were calculated. The first 500,000 iterations of the Markov chain were discarded as burn-in. For each fitted model, eight chains of parameters were sampled. Convergence of the chains was assessed through visual inspection of the trace plots and calculation of the Gelman-Rubin $\hat{R}$ statistic[63].

*Model selection.* Parameters were estimated for five epidemiological models:

- Model 1: constant transmission with different $\beta_i$ for each island,
- Model 2a: linear transmission with different $\alpha_i$ and the same $\beta$ for each island,
- Model 2b: linear transmission with the same $\alpha$ and different $\beta_i$ for each island,
- Model 3a: exponential transmission with different $\alpha_i$ and the same $\beta$ for each island, and
- Model 3b: exponential transmission with the same $\alpha$ and different $\beta_i$ for each island.

The best model was determined by which one had the lowest deviance information criterion (DIC)[21].

*Robustness of model fitting.* In order to confirm that the parameters to be estimated, $\theta$, were robustly inferred by our model fitting approach, we fitted our Model 3b to 10 synthetic data sets. Each synthetic data set was generated using the following procedure:

1. Each parameter was randomly sampled from their respective prior (Supplementary Table 4).
2. These parameters were collectively used to simulate the metapopulation model forward in time.
3. For each serological survey in the empirical data, we used the proportion of the population that were immune at time of testing (from the metapopulation model) to randomly sample the number of animals which tested RVF antibody positive using our observation model.

After fitting to each synthetic data, posterior distribution of parameters were compared to the values parameter values used to generate the synthetic data.

Posterior distributions for parameters estimated in each synthetic data set are presented in Supplementary Figs. 14–23. These figures are ordered from synthetic data sets with the most number of livestock infections (Supplementary Fig. 14), to the least number of livestock infections (Supplementary Fig. 23) during 2004–2015. All posterior distributions contained the parameter values used to generate the synthetic data. The data were informative over the priors for transmission constants, $b_i$, the NDVI transmission coefficient, $a$, and initial seroprevalence on each island, $\epsilon_i$, provided RVF outbreaks occurred in the synthetic data.

*Computational implementation.* The metapopulation model and Bayesian model fitting algorithms were coded and executed in C++14 with the GNU Scientific Library (version 2.6). Outputs from these algorithms were analysed and visualised in R (version 3.6.3)[64] using the tidyverse library (version 1.3.0)[65].

**Forecasting and control scenarios.** To assess the ability for RVF to persist within the Comoros archipelago beyond 2015, we forecast RVF virus seroprevalence in livestock using empirical NDVI data from 2015 until 2020. We did this by drawing 1000 samples from the joint posterior distribution of the best model fit and using these parameters to simulate the model until July 2020. To assess the importance of inter-island trade on the transmission of RVF within the archipelago, we imposed (100%) movement restrictions (imports and exports) on each island independently. To investigate the potential long-term impacts of island-specific control measures, such as vector control, on the transmission of RVF within the archipelago, we reduced the transmission rate by 10%, 20% and 30% on each island independently. For each control scenario, we calculated the total number of infections in livestock on each island compared with the total number of infections on the full movement and transmission network.

**Reporting summary.** Further information on research design is available in the Nature Research Reporting Summary linked to this article.

## Data availability
Summarised data used in our study to estimate the parameters in our metapopulation model, alongside a full description of the data, are available through the GitHub repository: wtennant/rvf_comoros[66]. These data are fully accessible and are presented in Fig. 2 and Supplementary Figs. 1–4.

## Code availability
All code, including the metapopulation model and fitting algorithm, are publicly available through the GitHub repository: wtennant/rvf_comoros[66].

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

## Acknowledgements

We thank Matthieu Roger, Floriane Boucher, the veterinary services and the surveyors for their help in collecting livestock trade data in the Union of the Comoros. We thank the veterinary services of Mayotte for sharing their surveillance data. This study was partly supported by the TROI project (FEDER Interreg V) under the DP One health Indian Ocean (www.onehealth-oi.org) in partnership. The study was conducted under

the Vaccine Efficacy Evaluation for Priority Emerging Diseases (VEEPED) project. W.S.D.T., W.J.E., M.J.K., M.J.T. and S.E.F.S. are funded by the Department of Health and Social Care using UK Aid funding managed by the National Institute for Health Research (Vaccine Efficacy Evaluation for Priority Emerging Diseases: PR-OD-1017-20007). The views expressed in this publication are those of the authors and not necessarily those of the Department of Health and Social Care. SEFS also greatly acknowledges support from from the Medical Research Council (MR/P026400/1) and Engineering and Physical Sciences Research Council (EP/R018561/1). M.J.T. is also funded by the Biotechnology and Biological Sciences Research Council (BB/T004312/1), further acknowledges funding, alongside M.J.K., from the Engineering and Physical Sciences Research Council through the MathSys CDT (EP/S022244/1). M.J.K. is also grateful for support from Health Data Research UK, which is funded by the UK Medical Research Council, Engineering and Physical Sciences Research Council, Economic and Social Research Council, Department of Health and Social Care (England), Chief Scientist Office of the Scottish Government Health and Social Care Directorates, Health and Social Care Research and Development Division (Welsh Government), Public Health Agency (Northern Ireland), British Heart Foundation and the Wellcome Trust.

## Author contributions

W.S.D.T., E.C., W.J.E. and R.M. conceptualised the study design. W.S.D.T., D.C., S.E.F.S. and R.M. designed the models and fitting algorithm, and W.S.D.T. implemented the code for the models and model fitting algorithm. E.C., C.C.S., Y.M., G.L.G. and O.C. acquired the data. W.S.D.T. performed the formal analysis of model results, and W.S.D.T., E.C., C.C.S., S.E.F.S., M.J.T., M.J.K., V.C., W.J.E. and R.M. interpreted the results. W.S.D.T. and R.M. prepared the first draft of the manuscript, and all authors critically reviewed and edited the manuscript. All authors approved the final draft of the manuscript.

## Competing interests

The authors declare no competing interests.
