## [Peer Review File · Nature Communications]

REVIEWER COMMENTS

Reviewer #1 (Remarks to the Author):

In this paper, a mathematical model is developed to describe the epidemiological dynamics of Rift Valley fever within the four islands of the Comoros archipelago. The model is parameterized using a Bayesian framework and data on age specific seroprevalence collected between 2004-2016 from the island of Mayotte and in 2009, 2012, 2013 and 2015 for the other three islands. The stated goal of the work was to better understand the spread and persistence of Rift Valley fever in this heterogeneous system and evaluate the impact of potential interventions. Overall, I think this paper addresses an important and interesting topic and uses a methodology that is, in the broadest sense, sound. At the same time, however, I was left disappointed by the absence of validation using simulated data and frustrated by the incomplete description of how the model and Bayesian analysis were performed. The generality of the work beyond Rift Valley fever is also somewhat unclear. In the paragraphs below, I detail these concerns and a few additional weaknesses that reduce the impact of the paper.

My greatest concern with this work is that results of analyses of simulated data are not reported. Consequently, it is unclear which parameters are identifiable given the structure of the data and the model. The accuracy of parameters estimates is also unclear absent any analysis of simulated data sets where the true values of parameters used to simulate the data are compared to those estimated by the authors' Bayesian approach. I recognize that the authors compared the fit of the model predictions to the data, but this does not yield the same information as testing against simulated data. I see this is a significant weakness of this work that makes interpretation of the results challenging.

Another piece of this work I found frustrating was the absence of a clear description of how the model and Bayesian analysis were integrated. Specifically, how are the time dependent simulations of the model coupled to the Bayesian approach? I assume this involved repeatedly simulating forward in time for specific parameter combinations to calculate the likelihood of the data over the time series, but this needs to be spelled out in much greater detail as it is not at all obvious how it was done.

The models upon which the work is based are deterministic. This makes discussion of pathogen persistence strange given that pathogen extinction is not possible within this deterministic framework. If the goal is to study mechanisms of persistence, a stochastic mathematical framework would be preferable, such as using the Gillespie algorithm to simulate continuous time ODE's. Taking this stochastic modeling approach would also increase the relevance of the model predictions which currently include error associated with random draws of the parameters from the posterior distribution but do not include process error associated with small numbers of, say, infected individuals.

This is a minor point but referring to credible intervals for predictions of IgG seroprevalence (Figure 4 legend) is confusing as IgG is a model output/prediction and not a model parameter with a posterior distribution.

Reviewer #2 (Remarks to the Author):

This is a clear and well written manuscript which presents a model fitted to surveillance data focused on Rift Valley Fever transmission in the Comoros. The Island nature of the setting allows a very neat metapopulation modelling framework to be applied and explore islands individually and as a group, connected through livestock movements, in the generation of outbreaks of RVF.

The objectives and approach are clearly laid out. The data were all secondary data, some of which are already published.

Two main methodological points stand out that deserve some comment in the manuscript.

1) NDVI is an important driving factor in the risk of RVF, yet NDVI data were summarised for all pixels on the island to produce one value per island per time point. I accept that the metapopulation patches are treated as one unit in this model, but wonder if a greater degree of resolution could have been obtained by modelling each island in more detail, with data on NDVI, livestock distribution on the islands etc more explicitly representing the reality on the ground.

2) Livestock movements, while in the end not the driving force behind the epidemiology, were summaries. It is likely that livestock movements follow some seasonal patterns and that accounting for this variability (where peaks may or may not co-incide with other risk factors like NDVI, for example) would also provide a level temporal refinement to the model proposed.

Otherwise, this is a neat and useful eco-epidemiological analysis.

A final thought is that the authors might go a bit further in helping the reader understand how this advances a broader epidemiological pattern away from this setting.....some more general thoughts about understanding disease epidemiology through an ecological approach such as this would be helpful.

Drivers of Rift Valley fever virus persistence and the impact of control measures in a spatially heterogeneous landscape: the case of the Comoros archipelago, 2004–2015

Response to reviewers' comments

We would like to thank the reviewers for their valuable feedback, comments and suggestions. We have carefully revised our manuscript taking care to address all concerns raised. In general, we have broadened our discussion suggesting how the results from our study might be applied to other RVF-affected regions, and how the metapopulation framework may be extended or adapted to other diseases circulating across multi-insular systems. Below, we address specific comments by each reviewer.

Reviewer #1

In this paper, a mathematical model is developed to describe the epidemiological dynamics of Rift Valley fever within the four islands of the Comoros archipelago. The model is parameterized using a Bayesian framework and data on age specific seroprevalence collected between 2004-2016 from the island of Mayotte and in 2009, 2012, 2013 and 2015 for the other three islands. The stated goal of the work was to better understand the spread and persistence of Rift Valley fever in this heterogeneous system and evaluate the impact of potential interventions. Overall, I think this paper addresses an important and interesting topic and uses a methodology that is, in the broadest sense, sound. At the same time, however, I was left disappointed by the absence of validation using simulated data and frustrated by the incomplete description of how the model and Bayesian analysis were performed. The generality of the work beyond Rift Valley fever is also somewhat unclear. In the paragraphs below, I detail these concerns and a few additional weaknesses that reduce the impact of the paper.

Thank you very much for your comments. We have now included the validation of our approach using simulated data, and provided a more detailed description of how the model and Bayesian analysis were performed. We have responded to these in more detail under the respective specific comments below.

With regards the generality of the work beyond Rift Valley fever: beyond Rift Valley fever, our modelling framework may be adapted to other diseases previously recorded to circulate in the Comoros archipelago. Our livestock movement estimates could be used to describe the spread of other livestock infections in the Comoros, such as Q-fever, *Peste des Petits Ruminants*, East-coast fever or Foot-and-Mouth disease (Dupont *et al.*, 1995; De Deken *et al.*, 2007; Cêtre-Sossah *et al.*, 2016). Our approach to modelling mosquito-borne diseases in the absence of entomological data could also be

adapted to describe other arboviral diseases. Furthermore, we have shown that our approach may be viable to estimate host movement patterns in multi-insular systems in cases where prior knowledge of movement patterns are uninformative (Supplementary Figure 13). Our framework could thus be adapted to infer host movement patterns using serological data.

We have added a paragraph in our discussion on the generality of the work within and beyond Rift Valley fever (lines 282–297).

De Deken R, Martin V, Saido A, Madder M, Brandt J, et al. (2007) An outbreak of East Coast Fever on the Comoros: A consequence of the import of immunised cattle from Tanzania? *Veterinary Parasitology* 143: 245-253.

Dupont, H. T., Brouqui, P., Faugere, B., & Raoult, D. (1995). Prevalence of antibodies to *Coxiella burnetii*, *Rickettsia conorii*, and *Rickettsia typhi* in seven African countries. *Clinical infectious diseases*, 21(5), 1126-1133.

Cêtre-Sossah, C., Kwiatek, O., Faharoudine, A., Soulé, M., Moutroifi, Y. O., Vrel, M. A., ... & Cardinale, E. (2016). Impact and epidemiological investigations into the incursion and spread of peste des petits ruminants in the Comoros Archipelago: an increased threat to surrounding Islands. *Transboundary and emerging diseases*, 63(4), 452-459.

My greatest concern with this work is that results of analyses of simulated data are not reported. Consequently, it is unclear which parameters are identifiable given the structure of the data and the model. The accuracy of parameters estimates is also unclear absent any analysis of simulated data sets where the true values of parameters used to simulate the data are compared to those estimated by the authors' Bayesian approach. I recognize that the authors compared the fit of the model predictions to the data, but this does not yield the same information as testing against simulated data. I see this is a significant weakness of this work that makes interpretation of the results challenging.

Thank you for raising this concern, this type of test was part of our initial methodology but was not initially reported to keep the manuscript more compact. We analysed the robustness of our model inference by fitting our model to 10 synthetic data sets. In order to generate each synthetic data:

- 1) we randomly sampled parameter values from their respective priors,
- 2) used these parameters to simulate our epidemiological model forward in time,
- 3) at each collection time in the empirical data, randomly sampled the number of livestock which would have tested positive for IgG-specific antibodies.

We then fitted our model back to each synthetic data set and compared the posterior distributions of parameters with the true values used to generate the synthetic data.

By analysing these posterior distributions, we concluded that parameters of interest were reliably inferred with our approach. That is, posterior distributions contained the parameter values used to

generate the synthetic data. This type of data was informative over-and-above the uninformative priors for transmission constants, NDVI-dependent transmission, and initial seroprevalence provided outbreaks of disease occurred in the synthetic data. In our study, we used informative priors for inter-island movement estimates, based consultation with Comorian veterinary services, and for importation timing and durations. We thus did not expect the data to be informative over these prior.

We have now included these details in the Methods and Materials section of the main manuscript (lines 438–455). We have also included figures showing the posterior distributions of inferred parameters against the true values used to generate each synthetic data (Supplementary Figures 14 to 23).

Another piece of this work I found frustrating was the absence of a clear description of how the model and Bayesian analysis were integrated. Specifically, how are the time dependent simulations of the model coupled to the Bayesian approach? I assume this involved repeatedly simulating forward in time for specific parameter combinations to calculate the likelihood of the data over the time series, but this needs to be spelled out in much greater detail as it is not at all obvious how it was done.

The link between the metapopulation model and Bayesian analysis was described using the observation model (simulating the model forwards in time and then using a binomial sample to capture the testing of animals). We have carefully reworded this section and added subsection titles to make this link more clear. We have also added a brief description of how the employed MCMC algorithm samples from the posterior distribution of parameters using the metapopulation model (lines 416–427).

The models upon which the work is based are deterministic. This makes discussion of pathogen persistence strange given that pathogen extinction is not possible within this deterministic framework. If the goal is to study mechanisms of persistence, a stochastic mathematical framework would be preferable, such as using the Gillespie algorithm to simulate continuous time ODE's. Taking this stochastic modeling approach would also increase the relevance of the model predictions which currently include error associated with random draws of the parameters from the posterior distribution but do not include process error associated with small numbers of, say, infected individuals.

Thank you for highlighting this issue; it was a question we had been thinking about previously given the relatively small size of the island populations. We acknowledged some limitations of using a discrete-time deterministic model in our Discussion (lines 256–265). Following the reviewer's remarks, we also developed a stochastic model, by simulating all transitions as probabilistic processes defined by Binomial probability distributions. This model formulation, however, requires additional input to explicitly describe the mechanisms of disease introduction (e.g. wildlife reservoirs of infection, importation of infectious livestock, or vertical transmission of the virus in vectors all require subtly different formulations). If these are present, we found that, with as little as one introduction per island per year, the stochastic model generated levels of persistence similar to the deterministic approach

outlined in the paper. Since we lack data to appropriately parametrise the precise introduction mechanisms required within the stochastic framework, we opted instead for using the deterministic model that corresponds to a more parsimonious modeling approach, allowing us to implicitly integrate these mechanisms and discuss persistence in terms of the reproduction number.

This showed that the disease could continue to propagate without an explicit large introduction of infections from continental Africa after 2007. However, we acknowledge that the aforementioned introduction mechanisms (which could implicitly allow for pathogen persistence within our framework) were not discussed in the original manuscript. These mechanisms may include vertical transmission of the Rift Valley fever virus in mosquito vectors (Manore *et al.*, 2015), availability of alternative hosts (Olive *et al.*, 2012) and irregular introductions of a small number of infected livestock through global trade (Fevre *et al.*, 2006). We have now highlighted these possible persistence mechanisms in our discussion (lines 266–269).

For the benefit of the reviewer, we include an example of output from our stochastic simulation model on the following page (Figure 1). These show that a stochastic model (with carefully chosen introduction rates) can generate results that are in broad agreement with the deterministic model results. However, due to the considerable uncertainty in the aforementioned mechanisms of introduction, and due to the far greater computational demands in fitting such stochastic models to data, considering the deterministic model only was the most appropriate decision for our research.

Manore, C. A., & Beechler, B. R. (2015). Inter-epidemic and between-season persistence of Rift Valley fever: Vertical transmission or cryptic cycling?. *Transboundary and emerging diseases*, 62(1), 13-23.

Olive, M. M., Goodman, S. M., & Reynes, J. M. (2012). The role of wild mammals in the maintenance of Rift Valley fever virus. *Journal of wildlife diseases*, 48(2), 241-266.

Fèvre, E. M., Bronsvoort, B. M. D. C., Hamilton, K. A., & Cleaveland, S. (2006). Animal movements and the spread of infectious diseases. *Trends in microbiology*, 14(3), 125-131.

This is a minor point but referring to credible intervals for predictions of IgG seroprevalence (Figure 4 legend) is confusing as IgG is a model output/prediction and not a model parameter with a posterior distribution.

Thank you for pointing this out. We now refer to intervals of IgG seroprevalence predictions as prediction intervals in both the main manuscript and supplementary figures.

Figure 1: Simulated seroprevalence in the stochastic model with introduction. We included an introduction term into the developed stochastic model using a Poisson distribution with rate of one introduction per year per island. Shown is the median (solid) and prediction intervals (PrI) of (bands) IgG seroprevalence in animals across all age groups in the stochastic model with one introduction per year per island on average. For reference, the median IgG seroprevalence in the deterministic model is shown (two-dashed). Summary statistics were calculated from 1,000 realisations of each model.

Reviewer #2

This is a clear and well written manuscript which presents a model fitted to surveillance data focussed on Rift Valley Fever transmission in the Comoros. The Island nature of the setting allows a very neat metapopulation modelling framework to be applied and explore islands individually and as a group, connected through livestock movements, in the generation of outbreaks of RVF.

The objectives and approach are clearly laid out. The data were all secondary data, some of which are already published.

Two main methodological points stand out that deserve some comment in the manuscript.

1) NDVI is an important driving factor in the risk of RVF, yet NDVI data were summarised for all pixels on the island to produce one value per island per time point. I accept that the metapopulation patches are treated as one unit in this model, but wonder if a greater degree of resolution could have been obtained by modelling each island in more detail, with data on NDVI, livestock distribution on the islands etc more explicitly representing the reality on the ground.

Thank you for your comments. We did not account for within-island spatial heterogeneity of RVF infection risk because disease data available for fitting our model at this scale were not available. There was also no information on the spatial distribution of livestock. We therefore assumed homogeneous mixing between animals within each island.

The relatively limited size of these islands—Grande Comore (1,146 km²), Moheli (290 km²), Anjouan (424 km²) and Mayotte (374 km²)—alongside the similarity of the ecosystem within each island (Le Goff *et al.*, 2014; Marsden *et al.*, 2013) and that RVF virus can be transmitted in the archipelago by a large diversity of mosquito vectors (Roger *et al.*, 2014; Metras *et al.*, 2020), supports that our assumption remains realistic.

We have now highlighted in our Discussion that our model could be extended to have more spatially-refined populations of livestock provided that the appropriate data streams are available (lines 288–292).

Le Goff, G., Goodman, S. M., Elguero, E., & Robert, V. (2014). Survey of the mosquitoes (Diptera: Culicidae) of Mayotte. *PLoS One*, 9(7), e100696.

Marsden, C. D., Cornel, A., Lee, Y., Sanford, M. R., Norris, L. C., Goodell, P. B., ... & Lanzaro, G. C. (2013). An analysis of two island groups as potential sites for trials of transgenic mosquitoes for malaria control. *Evolutionary applications*, 6(4), 706-720.

Roger, M., Beral, M., Licciardi, S., Soule, M., Faharoudine, A., Foray, C., ... & Cardinale, E. (2014). Evidence for circulation of the rift valley fever virus among livestock in the union of Comoros. *PLoS Negl Trop Dis*, 8(7), e3045.

Métras, R., Edmunds, W. J., Youssouffi, C., Dommergues, L., Fournié, G., Camacho, A., ... & Subiros, M. (2020). Estimation of Rift Valley fever virus spillover to humans during the Mayotte 2018–2019 epidemic. *Proceedings of the National Academy of Sciences*, 117(39), 24567-24574.

2) Livestock movements, while in the end not the driving force behind the epidemiology, were summaries. It is likely that livestock movements follow some seasonal patterns and that accounting for this variability (where peaks may or may not co-occur with other risk factors like NDVI, for example) would also provide a level temporal refinement to the model proposed.

Livestock movements across the Comoros archipelago have only been previously described qualitatively (Cêtre-Sossah *et al.*, 2012; Roger *et al.*, 2014; Métras *et al.*, 2017). These movements are a combination of legal and illegal trade and may depend on several factors including climate (season), geopolitical and economic reasons (e.g. price of an animal across islands), the timing of religious festivities, weddings or other family gatherings. In the absence of data in such complexity we assumed movements were constant year round. We agree that tracking animal movements would be very important for disease surveillance. We have now commented on this in our Discussion (lines 270–276).

Cêtre-Sossah, C., Pédarrieu, A., Guis, H., Defernez, C., Bouloy, M., Favre, J., ... & Albina, E. (2012). Prevalence of Rift Valley fever among ruminants, Mayotte. *Emerging infectious diseases*, 18(6), 972.

Roger, M., Beral, M., Licciardi, S., Soule, M., Faharoudine, A., Foray, C., ... & Cardinale, E. (2014). Evidence for circulation of the rift valley fever virus among livestock in the union of Comoros. *PLoS Negl Trop Dis*, 8(7), e3045.

Métras, R., Fournié, G., Dommergues, L., Camacho, A., Cavalerie, L., Mérot, P., ... & Edmunds, W. J. (2017). Drivers for Rift Valley fever emergence in Mayotte: A Bayesian modelling approach. *PLoS neglected tropical diseases*, 11(7), e0005767.

A final thought is that the authors might go a bit further in helping the reader understand how this advances a broader epidemiological pattern away from this setting.....some more general thoughts about understanding disease epidemiology through an ecological approach such as this would be helpful.

Thank you for this comment. We agree that we originally overlooked the broader application of our work. Some of this is partly addressed in our response to Reviewer #1.

The transmission parameters estimated by our framework, could be used to serve as initial input parameters to calibrate a RVF metapopulation model applied to other affected regions. We have shown that our approach may be used to estimate host movement patterns in multi-insular systems in cases where prior knowledge of movement patterns are uninformative (Supplementary Figure 13). Our framework could thus be applied to other diseases found in multi-insular systems, to estimate movement networks of bats or birds, for example (Tortosa *et al.*, 2012).

We have now added a section at the end of the Discussion, suggesting how our model, and the inferred parameters, may be extended and/or applied within and outside of RVF (lines 282–297).

Tortosa, P., Pascalis, H., Guernier, V., Cardinale, E., Le Corre, M., Goodman, S. M., & Dellagi, K. (2012). Deciphering arboviral emergence within insular ecosystems. *Infection, Genetics and Evolution*, 12(6), 1333-1339.

REVIEWERS' COMMENTS

Reviewer #2 (Remarks to the Author):

Thank you for your comments addressing the queries raised by the reviewers. I am now satisfied that the manuscript is of sufficient quality for publication.

Reviewer #3 (Remarks to the Author):

This manuscript has already been through one round of review. As one of the original reviewers was unavailable, I was asked to look at the authors' responses to the comments.

I read the manuscript with interest and I totally agree with the comments raised by Reviewer 1. The authors clearly took their feedback into account, and in my view, they adequately addressed all the comments. I feel reassured by the extra analysis the authors did by fitting the model to their synthetic data sets.

I have noticed a couple of minor points, mainly a question of definition, that I think the authors should address. I hope that these comments will help in improving the manuscript.

- Equations in the "Metapopulation model". Terms like $\delta, \mu_i, \lambda_{t,j}$ are generically defined as rates. The word "rate" usually suggest time into the denominator, if this is the case, then the equations are no longer dimensionally correct (the left-hand side is a number and the right-hand side is a function of time). Also, in principle a rate can be larger than one which make the interpretation of terms like $(1 - \mu_i)$ problematic. To me, terms like $\delta, \mu_i, \lambda_{t,j}$ are time-dependent proportions, but it would be good if the authors explicitly spell out the definitions and ensure that the equations are dimensionally correct.

- Although the notation is self-explanatory, unless I miss something, I am not sure if the authors provide a definition of $N_{t,i,a}$ and how these are related to N_i .

-Lines 129 – 130. When it reads: "...exports into Grande Comore by 100% increased the number of infections in Grande Comore itself by 10% (95% CrI = [-4.0%, 18.5%])." There is a sign minus in the Credible Interval, unless this was a typo, I presume that a negative number means reduction. I think the authors need to mention the implications of a credible interval overlapping with zero.

Modelling the persistence and control of Rift Valley fever virus in a spatially heterogeneous landscape

Response to reviewers' comments

We would like to thank the reviewers for their valuable comments on our manuscript. We would also like to thank the editor for efficiently navigating around the unavailability of Reviewer #1 from the first round of reviews. Below we respond to the comments of each reviewer.

Reviewer #2

Thank you for your comments addressing the queries raised by the reviewers. I am now satisfied that the manuscript is of sufficient quality for publication.

We are delighted that you are satisfied by our revisions. Thank you for your recommendation.

Reviewer #3

This manuscript has already been through one round of review. As one of the original reviewers was unavailable, I was asked to look at the authors' responses to the comments.

I read the manuscript with interest and I totally agree with the comments raised by Reviewer 1. The authors clearly took their feedback into account, and in my view, they adequately addressed all the comments. I feel reassured by the extra analysis the authors did by fitting the model to their synthetic data sets.

I have noticed a couple of minor points, mainly a question of definition, that I think the authors should address. I hope that these comments will help in improving the manuscript.

Thank you for taking the time to review our article. We appreciate that you took our response to Reviewer #1's comments on board. Below we respond to your minor points.

Equations in the "Metapopulation model". Terms like $\delta, \mu_i, \lambda_{t,j}$ are generically defined as rates. The word "rate" usually suggest time into the denominator, if this is the case, then the equations are no longer dimensionally correct (the left-hand side is a number and the right-hand side is a function of time). Also, in principle a rate can be larger than one which make the interpretation of terms like $\left(1 - \mu_i\right)$ problematic. To me, terms like $\delta, \mu_i, \lambda_{t,j}$ are time-dependent proportions,

but it would be good if the authors explicitly spell out the definitions and ensure that the equations are dimensionally correct.

You are correct; thank you. The terms, δ , μ_i and λ_{t_j} were all time-dependent proportions. That is, they are the proportion of the population which age, are removed from the system (given their age) and are infected at time t on island j at each time step (one week), respectively. We have clarified these units in our revised manuscript (lines 327–328, 330–331, & 337–338, and Table 1).

Although the notation is self-explanatory, unless I miss something, I am not sure if the authors provide a definition of $N_{\{t,i,a\}}$ and how these are related to N_i .

N_i referred to the total number of animals on island i , whilst $N_{\{t, i, a\}}$ refers the the number of animals in age group a on island i at time t . The total number of animals on each island, N_i , is equal to the sum of $N_{\{t, i, a\}}$ across all age groups. We have now stated this in the manuscript (lines 326–327, & 340).

Lines 129 – 130. When it reads: “..exports into Grande Comore by 100% increased the number of infections in Grande Comore itself by 10% (95% CrI = [-4.0%, 18.5%]).” There is a sign minus in the Credible Interval, unless this was a typo, I presume that a negative number means reduction. I think the authors need to mention the implications of a credible interval overlapping with zero.

The minus sign is supposed to be there in the credible interval. This means that there is a possibility (probability of greater than 5%) for the control measure to be effective at reducing disease burden in Grande Comore. This would mean that we cannot conclusively say that the control measure will increase overall disease burden (although we can say it is more likely to increase than decrease). We have now stated this in our manuscript (line 119, lines 234–239).